# Insights Gained in the Aftermath of the COVID-19 Pandemic: A Follow-Up Survey of a Recreational Training Program, Focusing on Sense of Coherence and Sleep Quality

**DOI:** 10.3390/ijerph17249201

**Published:** 2020-12-09

**Authors:** Etelka Szovák, Károly Varga, Imre Zoltán Pelyva, Rita Soós, Sára Jeges, Zsuzsanna Kívés, Ákos Levente Tóth

**Affiliations:** 1Doctoral School of Health Sciences, Faculty of Health Sciences, University of Pécs, H-7621 Pécs, Hungary; pelyva.imre@gmail.com (I.Z.P.); soosrita8@gmail.com (R.S.); jegessara@gmail.com (S.J.); 2Sociological Institute, Pázmány Péter Catholic University, H-2087 Piliscsaba, Hungary; var9184@iif.hu; 3Institute for Health Insurance, Faculty of Health Sciences, University of Pécs, H-7621 Pécs, Hungary; zsuzsa.kives@etk.pte.hu; 4Institute of Sport Sciences and Physical Education, Faculty of Science, University of Pécs, H-7624 Pécs, Hungary; tothahu@gmail.com

**Keywords:** physical activity, COVID-19 pandemic, sense of coherence, sleep quality, follow-up research

## Abstract

The original aim of this study was a follow-up assessment of a recreational program running for six months (September 2019–February 2020) within controlled conditions. Following the arrival of the COVID-19 pandemic, the survey acquired a new goal: how do the subjects of the follow-up sampling experience this severe stress situation, and in this experience, what role does physical activity and a salutogenetic sense of coherence play. Austrian women (*N* = 53) took part in the training program, whose physical condition was assessed before the start of the program, then reassessed after three months and after six months; the organizers also had them fill out the sense of coherence questionnaire (SOC) as well as the Regensburger insomnia scale. After the lifting of the lockdown introduced due to the pandemic, participants completed an online survey relating to their changed life conditions, physical activities, sense of coherence and sleep quality. Results: After the first three months of the training, no significant changes were detected. After six months, the participants SOC and sleep quality improved (Friedman test: *p* = 0.005 and *p* < 0.001). During the lockdown, sleep quality generally deteriorated (W-rank test: *p* = 0.001), while SOC did not change. The women in possession of a relatively stronger SOC continued the training (OR = 3.6, CI 95% = 1.2–12.2), and their sleep quality deteriorated to a lesser degree. (OR = 1.7, CI 95% = 1.1–2.8). Conclusion: The data reinforce the interdependency between physical exercise (PE) and SOC; furthermore, the personal training that the authors formulated for middle-aged women proved to be successful in strengthening their sense of coherence, and it also reduced the deterioration in sleep quality due to stress.

## 1. Introduction

Physical inactivity is one of the worldwide risk factors of non-infectious diseases, imposing a serious burden on society [1,2,3]. A higher physical activity correlates with lower mortality rates, in both sexes and in younger as well as older age groups. Numerous studies corroborate the beneficial effects of physical activity on mental and physical health at any age [1,4,5,6,7]. According to clinical trials, physical activity-related interventions promote health processes, as well as helping maintain independence for the elderly population [8,9,10].

The worldwide spread of COVID-19 has upset the normality of daily life, forcing the population to social distancing and self-isolation. Since these, the containment precautions also concern physical activities; home workouts remained the only possibility for playing to play sports and staying stay active during the pandemic. Total physical activity significantly decreased as a result of the between before and during COVID-19 pandemic in all age groups and especially in men [11,12]. A significant positive correlation was found between the variation of physical activity and mental well-being, suggesting that the reduction of total physical activity had a profoundly negative impact on the psychological health and well-being of the population [11].

Within the research of factors motivating physical activity, exploring the connection between physical activity and the “salutogenetic sense of coherence” is an emerging trend of the last two decades. Salutogenesis is a stress resource-oriented concept, which focuses on assets, strengths, and motivation as a way to maintain and improve the movement toward health. The core constructs salutogenic model is the sense of coherence scale (SOC), which has three components: comprehensibility, manageability, and meaningfulness [13,14].

People with a strong SOC feel like they understand the phenomena at work around and within them better, are more capable of managing them, and see the sense of their actions. All these components mobilize their generalized resistance resources (GRR) [15], which help combat illness. Shaped by his life work, numerous psychological phenomena (such as successfully coping with the stressors of everyday life or illnesses) acquired an explanation; up to this day, this theory is indicating the direction for the practice of health enhancement [13,16]. It was believed for a long time (even by Antonovsky himself) that the SOC is a stable entity that settles in the course of socialization by the age of 30 [17,18]. However, this thesis has been partially debunked; several studies demonstrated that within certain circumstances, SOC could change [19].

Earlier on, salutogenesis-related research was primarily directed at verifying the varying effects of SOC (e.g., the emergence of emotional and mental illness, the healing process for other diseases, etc.) as a predictor. For those with a strong SOC, coping with stress works better than people with weaker SOCs [20,21].

For a long time thought the researchers see the potential of strengthening SOC in the field of education, which can mainly be successful in childhood [22,23].

According to research among university students, more physical activity and more attention devoted to health and nutrition were positively associated with SOC [24,25,26,27].

Based on the technical literature, S. Super [17] demonstrated the mechanisms manifesting behind the complicated chain of connections between the internal and external circumstances that influence the enhancement and strengthening of the sense of coherence on one hand, and the effects and counter-effects of a strong sense of coherence on the other. A simplified model is provided below (Figure 1):

This figure reveals that shifting along the health-sickness continuum depends on the successful management of the “state of tension”, which is determined by SOC (arrows 4 and 5). Furthermore, successful tension management strengthens SOC, as well (arrow 3). Better health status is capable of better mobilizing general resistance resources (GRR) (arrow 6), which influence SOC through life experiences (arrow 7).

Insomnia is the condition of an inability or difficultly in starting and maintaining sleep, waking up early, and having an interrupted or nonrestorative sleep. Symptoms lasting from one to six months are regarded as subacute insomnia, whereas if these symptoms persist for longer than six months, they qualify as chronic insomnia. Stressful life events are closely related to the beginning of chronic insomnia [28].

Sleep quality has a critical role in promoting health since sleep disturbance has a powerful influence on the risk of depression [29]. In healthy adults, short-term consequences of sleep disruption include increased stress responsivity, somatic pain, reduced quality of life, emotional distress and mood disorders, cognitive, memory, and performance deficits” [30] (p. 151). “Long-term consequences of sleep disruption in otherwise healthy individuals include hypertension, dyslipidemia, cardiovascular disease, weight-related issues, metabolic syndrome, type 2 diabetes mellitus, and colorectal cancer. All-cause mortality is also increased in men with sleep disturbances” [30] (p. 151). Poor sleep quality is a key feature of insomnia [31].

The antecedent of the research covered by this study was a follow-up study in Hungary, conducted between 2008 and 2010 [32]. In that follow-up study, 33 men and 73 women, it was established that compared to the start of training, a positively oriented change occurred in the dimensions of mental, emotional, and social health as well as in all the dimensions of SOC and vegetative lability (measured with a Vegetative Lability Index) [33]. However, it is an open question, how the minimum duration required for the effects of the training to manifest, and furthermore, how enduring the achieved results would be.

The original aim of our research started in September 2019, was to disclose and demonstrate, in the richest possible detail, the effects of a physical recreational training program on body weight, fitness and SOC, and was planned to last for six months. The project was just about to be completed according to the research protocol when the COVID-19 pandemic took hold, halting personal contacts with the program’s participants. Due to the pandemic, we could no longer measure endurance beyond six months of the program; at the same time, we could analyze to what extent both the SOC and physical activity influence the reaction to a severe stress situation that affects everybody.

This unexpected occurrence presented an opportunity to define a new research goal, namely: examining how the research participants reacted to the pandemic and the lockdown that was implemented as a result. On one hand, did they continue with training according to the trainer’s remote guidance, or did they exercise significantly less than before the epidemic? On the other hand, did the strength of their salutogenetic sense of coherence have an influence on this lifestyle choice, and what was most important: did all of these circumstances affect their sleep quality, as an “imprint” of the effect of stress and a sensitive indicator of health condition.

Applying these experiences and adapting them to the needs of the given population, as part of a personal training involving two to three people, Etelka Szovák continued the program in Austria, and we designed the research project that constitutes the subject of the present article. Our aim was to investigate what physical and mental changes could be observed in women participating in the program and completing the training continuously and identically, first after three months, then after six months, controlling for other factors.

## 2. Materials and Methods

### 2.1. Participants and Procedure

Participants of this study were women living in rural areas of the Austrian province of Burgenland (Rechnitz, Schachendorf, Großpetersdorf, Schandorf, Dürnbach, Hannesdorf, Burg). Training sessions and survey administration were carried out at a personal training center at Schachendorf. The training itself was personalized, held in groups of two to three individuals, the effects of which on the participants’ physical and mental health were regularly checked by means of routine assessments.

From September 2019, the organizers chose volunteers to be in the sample for the research project; these individuals agreed to participate twice or three times weekly in the physical recreation training developed by Etelka Szovák, as well as lifestyle-related counseling; furthermore, they consented to their data being used anonymously for the purpose of statistical analysis. Another criterion for being selected for the sample was to have no active sporting past and to be not pursuing any other sports at the time of the training. During these six months, those who did not meet one or more of these conditions were excluded. Figure 2 illustrates the sample selection process.

Fifty-three women between the ages of 30 and 47 met the criteria, with whom we successfully established and maintained follow-ups until the present day. Their average age was 39.3 years (SD = 5.4), with 49% of them being younger than 40 years old. 80% of the women belonging to the under 40 years old have underaged children, while in the older age group, this percentage is 24%. As for their academic grades, the total sample does not differ significantly, with 33% of the participants having a maximum of a professional trade school grade, 41% having a medium grade (final exam), and 26% of them possessing a college or university diploma.

Prior to the lockdown, with the exception of two individuals who were stay-at-home mothers, all participants were in employment, with 71% of them working fewer than six hours a day. 53% of those employed had intellectual work, 35% did primarily physical work, and 12% had jobs that fit both descriptions. 77% had, on average, at least 2–3 h of free time per day, 43% had three or more.

Upon starting the program (2–7 September 2019), in the 12th week of the training (18–22 November 2019) and in the 19th week (10–14 February 2020), the participants filled out a paper-based questionnaire. Body size and fitness-related measurements were also recorded, with the research participants cooperating in the process (See Section 2.3, Measurement chapter).

Austria introduced restrictions due to the COVID-19 pandemic on 16 March 2020. For two weeks, a complete lockdown was implemented, which meant that leaving one’s residence was not allowed, not even for the purpose of purchasing food. Social care workers appointed by the local government delivered the food to everyone’s home. After this, a less restrictive lockdown took effect (with permission to leave one’s home only for dog walking and shopping once a day) until 1 May.

The fourth assessment was completed in the first week of May, in the form of a questionnaire administered online. Compared to the paper-based questionnaire, this questionnaire asked questions about activities during the pandemic, focusing on the shift in lifestyle and its subjective assessment; however, measurements, which would have required the assistance of the trainer or helpers, were not carried out.

### 2.2. Recreational Training

The training plan, which was implemented from September 2019, required the completion of a 60-min workout twice a week and a leisurely Iyengar yoga session once a week. Duration and difficulty level of tasks were personalized to participants. The primary aspects of determining these tasks were the participants’ age and actual fitness status. In the course of these training sessions, the heart rate, blood pressure, and observations about the participants’ fitness condition were measured and recorded; furthermore, the individuals’ perception of gradually intensified tasks was noted. Besides physical training, the trainer also provided healthcare and lifestyle counseling, in the course of which the acquired results were individually assessed. The participants also practiced elements of mindfulness-based stress-reducing training (MBSR) [34]. Since an important motivation for participating in the survey was, besides health preservation, to achieve an esthetically pleasing figure, the thematic was from the bases of classic fitness as well as body- and figure-shaping trends [35]. One of the most important added elements of these trends was control of heart rate. By means of a POLAR FT7 heart-rate monitor, the heart rate was kept in a 135–155 min/s fat-burning zone; that is, the heart rate (HRmax) is 60–70% of the maximal heart rate [36].

Toning primarily targeted the abdomen muscles, thigh muscles, glute-muscles and hip muscles, which the author named BOGH, based on the exercised body areas.

Training content: 1. Warmup with a spinning bicycle; 2. Targeted toning of muscle groups; 3. Stretching: exercises with a stretching effect. In the course of establishing the training content, the trainer endeavored to make the participants complete simply choreographed gymnastic exercise sets, keeping in mind that the target group did not like training contents requiring stronger endurance. The target was set to reduce the body’s fat tissue, applying combinations of steps without skipping, strengthening and stretching exercises. With smaller weights, the participants completed shoulder, chest, and back muscle toning. For strengthening the abdomen, hip, thigh and glute muscles, they worked out eight minutes on average per body area, and then they proceeded to do stretching exercises. The equipment used during trainings were: jogging machine, spinning bicycle, free weights, attachable leg and hand weights, TheraBand bands, TRX, fitness ball and kettlebell.

### 2.3. Measurements

The organizers recorded measurements regarding sociodemographic characteristics, lifestyle and health preservation, as well as the motivation to participate in the program before the program started. Throughout the program, the body size-related parameters such as body weight, height, chest width, hip width, waist circumference, right and left thigh circumference, and bicep circumference were measured, and the body fat percentage, as well as the Body Mass Index (BMI) index, was determined three times (before starting the training, after three months and after six months). The body fat percentage was calculated based on the skin ridge in three areas (waist, hip, neck), using the US Navy body fat formula for females [37].

Fitness was measured before starting the program and after six months of training, based on the following data: how many pushups, squats, and sit-ups the participant could produce per minute, and the stretchability of leg muscles was measured (that is: how far can the individual stretch her arms forward while sitting), as well as the resting blood pressure and heart rate. During all four surveys, Regensburger’s sleep quality test (RIS) was filled out, along with the six-item version of the sense of coherence scale (SOC-6), in the course of the first three assessments, and the 13-item version (SOC-13) during the last occasion. Furthermore, important lifestyle-related data were recorded, such as the rhythm of daily activities, eating habits and occasional extraordinary occurrences. Lifestyle information was recorded based on a self-compiled questionnaire. We applied the food frequency questionnaire and Yale food addiction scale studies to compile nutrition questions [38,39]. The fourth survey contained further questions related to the pandemic, such as to what extent the lockdown changed participants’ everyday activities and how much difficulty this caused them.

#### 2.3.1. Sense of Coherence

The items in the six-item version of the Sense of Coherence Scale were chosen, based on psychometric analyses, from the original 29 items, elaborated by Antonovsky [13]. A 7-point rating scale (mostly ranging from 1 = very rarely to 7 = very often) to measure regularity and experienced feelings were used to measure each item. “It often happens” indicated a weaker sense of coherence, while a higher total score would correspond to a stronger sense of coherence. The six-item SOC scale contains the following items:

Two items referring to comprehensibility: “Do you have very mixed-up feeling and ideas? Very often (1) to very seldom or never (7)”, and “Does it happen that you have feelings inside you would rather not feel? Very often (1) to very seldom or never (7)”.

Two items referred to manageability: “Do you have the feeling that you’re being treated unfairly? Very often (1) to very seldom or never (7)” and “When something unpleasant happened in the past your tendency was: to beat yourself up about it (1) to say ok, that’s that, I have to live with it and go on (7)”.

Two items referring to meaningfulness: “When you think about your life, you very often: feel how good it is to be alive (1) to ask yourself why you exist at all (7)”* and “You anticipate that your personal life in the future will be: totally without meaning or purpose (1) to full of meaning and purpose 7)”.

* Note: In the case of this item, when calculating the score sum, the coding direction was inverted (1 = 7, 2 = 6 … 7 = 1), so that answers suggesting a higher sense of coherence were scored higher.

In the 53-person sample appearing in our present study, the value of Cronbach’s alpha came out to be 0.74, 0.66 and 0.78 in three interviews. During the fourth survey, the Cronbach’s alpha value of the 13 item scale was 0.89.

#### 2.3.2. Regensburger Insomnia Scale

The Regensburger insomnia scale (RIS) was designed in German. In our survey, we applied a narrowed-down version of the original scale, which was validated by Tatjana Crönlein and associates, and published in 2013. The Regensburger insomnia scale (RIS) is a new self-rating scale to assess cognitive, emotional and behavioral aspects of psychophysiological insomnia (PI) with only ten items. Five items were selected to cover quantitative and qualitative sleep parameters: sleep latency (1), sleep duration (2), sleep continuity (3), early awakening (4) and sleep depth (5). Four items asked about the psychological aspects of PI, such as the experience of sleepless nights (6), focusing on sleep (7), fear of insomnia (8), and daytime fitness (9), one item was about sleep medication (10). A 5-step Likert scale was provided for response. This type of scale was also used for quantitative sleep parameters (sleep duration and sleep latency) because, according to our clinical experience, insomnia patients have difficulties giving exact answers when asked about quantitative data. The total score ranged from 0 to 40 points [40].

Scores of 0–12 points do not indicate a significant sleep problem, 13–24 points amount to a remarkable amount (“mild insomnia”), and a value between 25 and 40 already equals a PI.

### 2.4. Statistical Analysis

Statistical analyses were carried out using SPSS v. 25.0 (IBM Corp. Released 2017. IBM SPSS Statistics for Windows, Version 25.0, IBM Corp., Armonk, NY, USA). We applied a related samples Friedman nonparametric test for testing the values of the examined variables measured at three different points in time, as well as in-pair comparisons. We examined the results of the third survey before the pandemic and the results of the last survey with the related samples Wilcoxon signed-rank test and the McNemar test in the case of dichotomized variables. The application of nonparametric tests is justified by the fact that variables were either measurable on an ordinal scale or were quantitative; however, their distribution did not meet the conditions of the parametric tests.

Using factor analysis, we analyzed the extent of the motivation of the participants to consume healthy food. A binary logistic analysis was conducted with reported sleep problems (total score < 13 versus ≥ 13), as well as between environmental factors and SOC. For examining the independence of the dichotomy variables, we applied a chi-squared test and calculated odds ratios (OR). In surveys carried out before introducing lockdown measures, SOC was assessed with a six-item scale, while in post-lockdown surveys, SOC was assessed with a 13-item scale. For comparability, we, therefore, grouped individuals into strong and weak SOC categories (k-means cluster analysis, number of clusters = 2) using cluster analysis, and then, we compared the SOC “measured” with this cluster variable via the McNemar test (binomial distribution used).

* Note: In both cases, the clusters significantly differed from one another among the corresponding items (Mann–Whitney test, *p* < 0.010).

### 2.5. Ethics

Study protocols were in accordance with the latest version of the Declaration of Helsinki. The Institutional Review Board of the University of Pécs—Clinical Center, Regional and Institutional Research Ethical Committee approved the study (reference number: 7520-PTE 2018; the approval number of the survey supplemented by post-lockdown testing: 7520-PTE 2020). All participants were duly informed about the study, and all provided informed consent.

## 3. Results

### 3.1. Results of the Surveys Completed before the Pandemic, in the Course of the Recreational Training

#### 3.1.1. Changes in Body Size

In the Table 1, we present average values of parameters significant in terms of figure-shaping, as well as the body fat percentage and the average values of BMI, along with significance levels of the changes in measurements between months 0–3 and 3–6, respectively. After the first three months of training, the examined parameters did not change except for BMI, which reduced significantly. Between the third and sixth months of training, every parameter related to body shape changed favorably from the participants’ perspective, except for bicep circumference. However, BMI increased significantly and largely returned to the initial value.

#### 3.1.2. Change in Fitness Status

Table 2 shows the changes in participants’ fitness parameters from the beginning of training to six months later. Wilcoxon-rank tests suggest a significant improvement in every parameter.

#### 3.1.3. Eating Habits and Adherence to Diet-Related Advice

Regarding the consumption of certain types of foods (such as meat, fish, rice, bread, vegetable fruits, “snacks”, and so on), participants were asked at the end of the third and sixth months of the training whether they consumed more of these foods in the period in question than before the training and whether they adhered to the diet prescribed for them. According to the results gained through a principal component analysis and applying varimax rotation (Kaiser-Meyer-Olkin Measure of Sampling = 0.68, the significance level of the Bartlett test *p* < 0.001), in the bi-factor solution, one factor contained foods considered healthy, while the other was comprised of foods recommended for moderate consumption, with high factor loadings. We compared the factor values in terms of whether the respondent self-reportedly adhered to the diet. In the case of both factors, the difference was significant; that is, those who adhered to the diet did consume much healthier food during the training than previously.

The change in body fat percentage as well as the correlation and level of significance between preferring a “healthy” or “unhealthy” diet are shown in Table 3.

A difference was only detected after six months, and with preferring healthy foods associated with losing weight.

#### 3.1.4. Changes in the Sense of Coherence as Well as Sleep Quality

Table 4 shows changes in the three SOC dimensions and the total score of the insomnia scale. A similar phenomenon to that observed in the change of physical parameters is also observed in the change of SOC after completing the training program; that is, after three months, changes were not yet statistically significant, whereas, after six months, the change were statistically significant. However, sleeping problems significantly reduced after only three months.

### 3.2. Results of the Survey Related to the Lockdown Implemented Due to the Pandemic

#### 3.2.1. Life Circumstances

During the curfew restrictions, 60% of the employees continued their work in the “home office”. At the time of the lockdown, 45.3% of the interviewed women lived in a house with a large garden, and 30.2% lived in a house with a smaller garden. Altogether 19.3% did not live in a house with a garden but had a large park nearby or lived in a rural cottage. Based on these circumstances, the interviewed women (94.8%) were presumably less impacted by the state of reclusion than those who were forced to spend this period “within four walls”. Among the participants, 28.3% did not continue at all the exercises “learned” during the training before the pandemic—of course, referring to those exercises which did not require any gym equipment. Altogether 41.5% partially continued with the workouts, while 30.2% went on to exercise regularly. Regarding time spent outdoors, 53.6% reported being outdoors for about the same time or even more (than before). In the period after the two-week total lockdown, 41.5% left their homes at least once, while 34.0% of them did not go out at all.

#### 3.2.2. Sleep Quality Deterioration

In the entire sample, according to the related samples Wilcoxon signed-rank test, the total score of the RIS increased significantly (*p* = 0.001). Based on the sleep quality (RIS) total score values, we created a qualitative variable (RIS_dich), with the possible values being 0 if the individual had no remarkable sleep disorder and 1 if the total score was higher than 12; that is, if a mild or more serious sleep disorder occurred. (A severe sleep disorder, that is, psychophysiological insomnia, was not recorded for any participants before or after the start of the training program.) Immediately before the epidemic, four individuals (7.5%) had a “slightly bad” sleep quality. Among those who never had any sleep problems before, 19 (35.9%) now experienced sleep problems (McNemar test, *p* = 0.003).

#### 3.2.3. The Influence of Environmental Factors on Sleep Quality

Presumably, changes occurring in life circumstances influence sleep quality. However, after examining this with a multiple logistic regression model, the effect of these circumstances was impossible to determine (Table 5).

#### 3.2.4. The Effect of Physical Activity on Sleep Quality

Pre-pandemic surveys verified the positive influence of recreational training on sleep quality. The pandemic disrupted the continuity of the original training plan; however, for the individuals in the follow-up, the trainer continued to provide them with a training plan remotely. Although the trainer could not directly observe how these plans were implemented, it was possible to specify how much the participants’ physical activity changed compared to their previous activity. (Physical activity is 0 if the participant exercised less than previously; 1 if the participant continued with the training utilizing completing the exercises or in some other way.) Based on the chi-squared test, the sleep quality of those individuals who remained active during the pandemic was more likely not to deteriorate when compared to those who stopped the training and exercised less (*p* = 0.014; OR = 4.8; CI 95% = 1.3–17.3).

#### 3.2.5. Connection between the Sense of Coherence and Physical Activity, and Their Effect on Sleep Quality

Those who possessed a relatively stronger SOC the odds were 3–4 times greater to exercise during the pandemic (*p* = 0.034; OR = 3.6; CI 95% = 1.2–12.2). The likelihood of sleep quality not deteriorating for participants with a strong sense of coherence odds are 1,7 times greater than for those who have a weaker sense of coherence (*p* = 0.024; OR = 1.7; CI 95% = 1.1–2.8).

We also examined the likelihood of sleep quality deterioration (deteriorated = 1, did not deteriorate = 0) depending on the physical activity and SOC variables, including them together in a multiple logistic regression analysis. Table 6 suggests that in terms of SOC and physical activity—both influencing the sleep quality related to the pandemic—physical activity is the more important moderating factor since, in the model simultaneously controlling for both of these variables, physical activity remains significant (*p* < 0.050), while the significance of SOC disappears.

As a summary of the results, a simplified model can be established as below (Figure 3):

A relationship between the strength of the sense of coherence and sleep quality was shown for each measurement. Those with a stronger sense of coherence had fewer sleep problems. This is illustrated by the arrows 1–3.

Based on the results we gained in the first phase of the survey (in the course of the training before the pandemic):

After the first three months of the training, generally, no significant changes were observed in the measured parameters. (Therefore, RIS 3 and sense of coherence 3 are not shown in the figure).

However, after training for six months, the parameters measuring both physical and mental health shifted in a positive direction. Sleep quality improved, and SOC enhanced (arrows 4 and 5).

During the pandemic:

In the survey after the lockdown imposed due to the pandemic, there were generally more sleep problems than immediately after six months of training (arrow 6), while the SOC did not change to a statistically detectable degree (arrow 7).

Individuals with a relatively stronger SOC continued the training, or they were physically active in other ways, whereas those who had a weaker SOC exercised less than in the course of the training program (arrow 8). The sleep quality of those who were physically more active was less impaired than those who were inactive (arrow 9). Physical activity acted as a mediator in the relationship between SOC and sleep quality impairment (arrow 10).

## 4. Discussion

S. Super’s model [17] (simplified post-Antonovsky), as described in the introduction, highlights the importance of the core function of the individual’s sense of coherence in terms of coping with stress, and thus, preserving and developing health. From the model figure (Figure 1), we can deduce that at the same time, the sense of coherence is a dependent variable of health development; that is, due to genetic and other environmental GRR, the health of individuals with an already stronger sense of coherence can be further developed.

The aim of our survey before the pandemic designed for examining the effects of the recreational training program was, in a more generic phrasing, to check whether the personal training, completed according to the project, verified the above claims. Our data confirm the existence of an interdependent relationship between physical activity and the SOC. At the end of the first three months of the training, a statistically significant change was not detected; however, by the end of the sixth month, the participants’ SOC had strengthened significantly. Several previous studies have reported that physical activities (PA) may lead to an increase in SOC [41,42,43,44]. The previous study found a report that regardless of age, sex, nationality or ethnicity, adults with a strong SOC had been found to have lower diastolic blood pressure, serum triglycerides, heart rate at rest, and higher oxygen uptake capacity, whereas a low SOC was associated with related to mental and circulatory health problems [45]. A strong SOC was also is associated with higher levels of health-related behaviors such as PE and lower rates of cigarette smoking [4,46].

With the COVID-19 pandemic, the world faced an unexpected challenge. The pandemic is having a profound effect on all aspects of society, including mental health and physical health [47,48,49]. As a result of the lockdown as well as the curfew, personal relations with the survey’s subjects were temporarily interrupted. The question arose how the women within our study experienced the pandemic as severe stress, as well as the changes in life circumstances brought about by it. Would there be any factors that positively affected stress processing?

Prior to the pandemic, the purpose of this study was to measure the impact of the personalized recreational training, which showed that, by the end of the sixth month, all physical conditions except BMI had changed in a positive direction to a small extent. Body fat percentage decreased; however, BMI increased from baseline. This can probably be attributed to the fact that the body fat was replaced by increased muscle mass. The change in fitness parameters also showed a slight improvement.

During the training, the researchers continued to have a personal relationship with the participants of the study. When this was no longer possible due to lockdown measures, new survey questions, the way in which information was obtained, and the shift to the online collection of data were formulated in a very short period of time. This method of empirical studies is preferred by most studies, such as [50,51].

One of the main results of this study related to sleep quality, as measured on the “Regensburger insomnia scale”. On one hand, a previous study had found that physical activity was a significant factor influencing sleep quality while controlling for other factors [52]. It is also clear that insomnia is associated with mental disorders such as depression, anxiety and that environmental factors such as stress have a significant role in the development of these disorders [53]. In our study, a significant deterioration in sleep quality was detected by the end of the curfew, but this was less significant among those who had stronger SOCs and continued training. According to Cellini et al., the proportion of poor sleepers increased from 40.5% to 52.4% under the restriction. The increase in sleep difficulties was stronger for people with higher levels of depression, anxiety and stress symptomatology and associated with the feeling of elongation of time [54]. Moderate physical activity, defined as below 3.0–6.0 exercise metabolic rates (METs) (3.5–7 kcal/min), (CDC) seems to be more effective than vigorous activity in improving sleep quality in both young and old populations [55]. The moderate activity is defined as below 3.0–6.0 exercise metabolic rates (METs) (3.5–7 kcal/min) (CDC).

The strength of this study is that in utilizing the experiences gained from surveys with similar themes, the authors endeavored to design a survey as exact as possible, from every aspect (inclusion and exclusion criteria, the accuracy of interviews and processing, choice of statistical analysis). The research is a longitudinal survey, which allows for more accurate deductions. Although the number of individuals in the sample is relatively low, this allowed the same trainer to lead training sessions and evaluate the experience gained during counseling, providing for consistency in training and evaluation. A weakness of this study is that a control group who did not attend training but was similar to those who participated in the training was not available. In addition, the number of individuals in the sample was low, thus less suitable for multivariate analysis, and with only very weak reliability.

The appearance of this pandemic created a severely stressful but unique natural experiment not seen in the world in the last hundred-to-hundred and fifty years. Rapidly reacting to this situation, our study supported our hypothesis, according to which a relatively stronger sense of coherence is capable of inducing a more consistent behavior even in a situation of severe stress, and thus, can moderate the unfavorable effects of stress.

## 5. Conclusions

After perusing the scientific literature and observing the results of these surveys, it suggests that developing areas contributing to strengthening the sense of coherence must become one of the main directives of developmental health programs, and among these areas, physical activity has a significant role. For a female population that is not forced to engage in PE during their everyday life, due to their largely sedentary work, with their financial resources allowing for employing helpers to complete around-the-house work, and moreover, lacking a sufficient motivation for engaging in sports, a personalized recreational training along with parallel counseling could effectively strengthen both their physical activity and sense of coherence.

Supporting the interdependent relationship of physical activity and the sense of coherence with empirical data contributes to a complex approach to answer the question, “Why are some people physically active and others not?”

## Figures and Tables

**Figure 1 ijerph-17-09201-f001:**
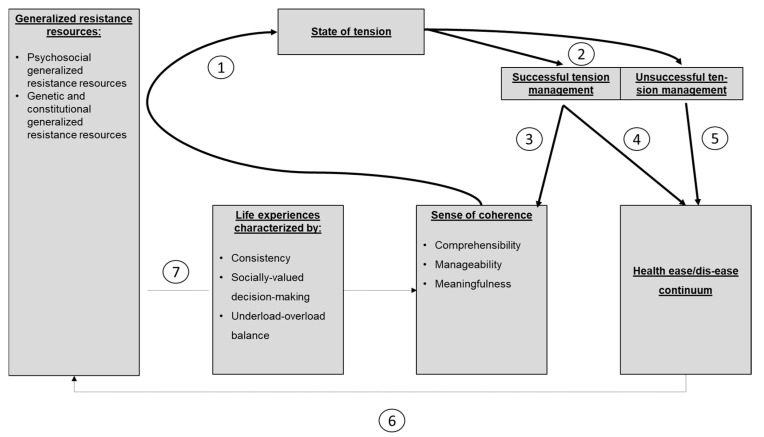
A simplified reproduction of the salutogenic model [17] (p. 871). (with the permission of the author).

**Figure 2 ijerph-17-09201-f002:**
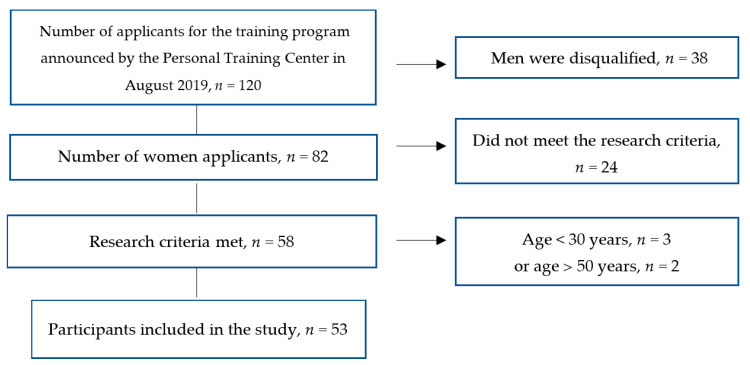
Flowchart showing the participant selection process.

**Figure 3 ijerph-17-09201-f003:**
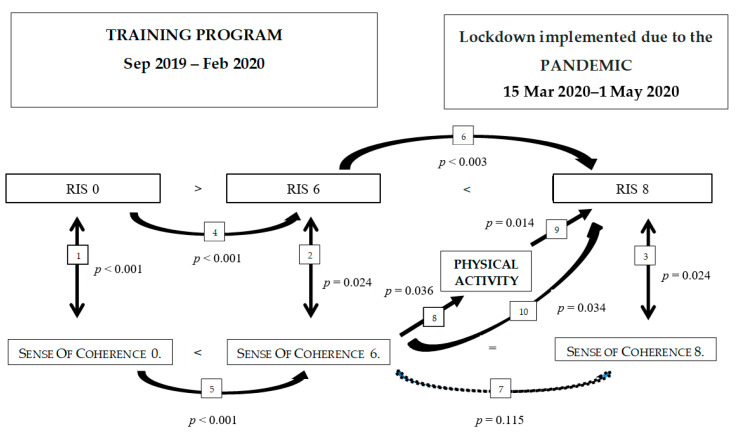
Physical activity and the effect of the sense of coherence before the training program and during the lockdown implemented due to the COVID-19 pandemic. RIS = Regensburger insomnia scale. The number displayed after the variable names is the surveys’ ordinal number: 0 = upon starting the training program, 6 = at the end of the training program (after the six months were completed), 8 = survey after the lockdown.

**Table 1 ijerph-17-09201-t001:** Body size means and standard deviation before starting the training, after three and after six months, as well as the significance levels of the Friedman test.

Anthropometric Characteristics	Survey Date	Total		Friedman Test Pairwise Comparisons Adj. Sig.		
		*N* = 53				
		Mean	Std.	0–3	3–6	0–6
Chest width (cm)	0: Before starting the training	94.5	8.9	0.808		0.001
	After the 3rd month	94.1	8.4		0.005	
	After the 6th month	93.3	8.0			
Hip width (cm)	0: Before starting the training	98.3	9.9	0.308		0.001
	After the 3rd month	97.2	9.1		<0.001	
	After the 6th month	96.0	8.5			
Waist circumference (cm)	0: Before starting the training	88.0	10.8	0.734		<0.001
	After the 3rd month	87.5	10.1		<0.001	
	After the 6th month	85.9	10.1			
Right thigh circumference (cm)	0: Before starting the training	55.3	4.7	0.884		<0.001
	After the 3rd month	55.1	4.6		<0.001	
	After the 6th month	54.0	4.4			
Biceps circumference (cm)	0: Before starting the training	30.0	3.6	0.627		0.321
	After the 3rd month	29.8	3.3		0.887	
	After the 6th month	29.9	4.9			
Body fat percentage	0: Before starting the training	28.5	5.9	0.285		<0.001
	After the 3rd month	27.9	5.3		<0.001	
	After the 6th month	26.9	3.3			
Body Mass Index (kg/m^2^)	0: Before starting the training	23.3	4.3	<0.001		0.627
	After the 3rd month	22.5	4.0		<0.001	
	After the 6th month	23.1	4.0			

**Table 2 ijerph-17-09201-t002:** Parameters describing fitness at months 0 and 6.

Exercises	Survey Date	Total		Wilcoxon-Rank Test Sig.
		*n* = 53		
		Mean	Std.	0 vs.6
Squats	0: Before starting the training	24.9	7.4	0.022
	After 6 months	26.5	6.9	
Pushups	0: Before starting the training	16.6	7.7	0.028
	After 6 months	17.5	7.4	
Sit-ups	0: Before starting the training	18.2	7.3	0.033
	After 6 months	19.9	6.0	
Stretches	0: Before starting the training	88.6	9.5	0.002
	After 6 months	89.1	10.1	

**Table 3 ijerph-17-09201-t003:** Correlation between consuming a healthy/unhealthy diet and the change in body fat percentage.

Surveys	Healthy Diet		Unhealthy Diet	
	Spearman’s Rho	Sig.	Spearman’s Rho	Sig.
Between months 0 and 3	−0.296	0.142	−0.201	0.315
Between months 3 and 6	−0.271	0.181	−0.195	0.331
Between months 0 and 6	−0.690	0.001	−0.202	0.312

**Table 4 ijerph-17-09201-t004:** Mean and standard deviation of sense of coherence traits, as well as the values measured on the Regensburger scale before starting the training, after three and after six months, as well as the significance levels of the Friedman test.

Variable	Survey Date	Total		Friedman Test Pairwise Comparisons Sig.		
		*N* = 53				
		Mean	Std.	0–3	3–6	0–6
Comprehensibility	0: Before starting the training	9.3	3.1	0.143		0.048
	After the 3rd month	9.8	2.6		0.142	
	After the 6th month	10.5	3.0			
Manageability	0: Before starting the training	8.7	2.5	0.174		0.003
	After the 3rd month	9.1	2.3		0.099	
	After the 6th month	10.1	3.1			
Meaningfulness	0: Before starting the training	11.3	2.0	0.145		0.029
	After the 3rd month	11.8	2.2		0.466	
	After the 6th month	12.1	2.3			
Sense of coherence	0: Before starting the training	29.3	5.8	0.356		0.005
	After the 3rd month	30.6	5.5		0.076	
	After the 6th month	32.7	7.6			
Regensburger insomnia scale	0: Before starting the training	16.1	3.1	<0.001		<0.001
	After the 3rd month	13.8	2.9		<0.001	
	After the 6th month	9.1	2.1			

**Table 5 ijerph-17-09201-t005:** The chance of deteriorating sleep quality in relation to changed life conditions as a result of the pandemic.

Variable	Sig.	Exp (B)	95% CI for EXP (B)	
			Lower	Upper
Do you live in a house with a garden? (yes = 1, no = 0)	0.152	0.307	0.061	1.544
Were you severely affected by the lockdown?	0.449	1.252	0.700	2.239
(yes = 1, no = 0)				
Did you spend less time outdoors?	0.842	1.064	0.578	1.957
(yes = 1, no = 0)				

**Table 6 ijerph-17-09201-t006:** Deterioration of sleep quality in relation to the sense of coherence and physical activity.

Variable	Sig.	Exp(B)	95% CI for EXP (B)	
			Lower	Upper
SOC (stronger = 0, weaker = 1)	0.091	2.101	2.101	0.887
Physical activity (yes = 0, no = 1)	0.036	0.383	0.383	0.156

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
