# Peer review of "Insights Gained in the Aftermath of the COVID-19 Pandemic: A Follow-Up Survey of a Recreational Training Program, Focusing on Sense of Coherence and Sleep Quality"

_ijerph, 2020, doi:10.3390/ijerph17249201_

Round 1

Reviewer 1 Report

Review ijerph-960280

First, I appreciate the opportunity to review this article. It is a current and necessary theme. The concept of Sense of Coherence is valuable and innovative. The report is very well written, with clarity and objectivity. The method is adequate and precise. As a contribution, I only request attention at some points in the study.

Introduction

Lines 40-41 –"A higher physical activity correlates with lower mortality rates, in both sexes and younger as well as older age groups."- This statement needs a reference.

Line 60 – "up to this day, his theory is indicating..." - It seems here that the word should be "this," that is, the letter "t" is missing. Please, check.

Figure1 - The direction of the arrow n. 7 has not been shown in the figure. According to the text, the path goes from general resistance resources to life experiences. Please, check.

Methods

Line 82 - "and 13 % "- Please, check it. 

Line 88 - "Personal conversations revealed that participants are …"- The verb tense in this section must be in the past. Please check.

Line 218 – "…furthermore, the individuals' reception..." - The verb tense in this section must be in the past. Please check.

 Please check if the word "reception" should be replaced by "perception."

Line 307 – "…the parameter tests."- Please, think about replacing "parameter" with "parametric."

Line 354 and title of the Table - Please, consider replacing the word "foods" with "diet" to keep the consistency and precision of the terms.

Line 374 – "by the state of seclusion." - Please verify the word "seclusion" and to replace it with "reclusion."

Figure 3 - Figure 3 is beautiful! Ingenious and very clear explanations! Congratulations! Just as a way to contribute, for this fantastic figure, I ask if it would be possible to make arrow no.10 pass through the box of physical activity to make the mediation of physical activity well designed {the variable that enters the chain causal and contributes to the outcome}. But that is just a note. As I said, the figure is very clever.

Discussion

- Very clear and precise. 

Line 482- "… measured on the Regensburger … "– Please, to check the signal that is highlighted in red/bold.

  1.  

Author Response

Response to Reviewer 1 Comments

First, I appreciate the opportunity to review this article. It is a current and necessary theme. The concept of Sense of Coherence is valuable and innovative. The report is very well written, with clarity and objectivity. The method is adequate and precise. As a contribution, I only request attention at some points in the study.

R: We would like to thank for the extensive review of this article. We have revised and modified the manuscript according to your suggestions.

Lines 40-41 –"A higher physical activity correlates with lower mortality rates, in both sexes and younger as well as older age groups."- This statement needs a reference.

R: The above finding is supported by the article published in Lancet 2012, 380, 219-229 [1], among others. The text has been supplemented by the reference [1].

Line 60 (=Line 68) – "up to this day, his theory is indicating..." - It seems here that the word should be "this," that is, the letter "t" is missing. Please, check.

R: Thank you, we have corrected.

Figure1 - The direction of the arrow n. 7 has not been shown in the figure. According to the text, the path goes from general resistance resources to life experiences. Please, check.

R: As the figure is a copy (quote) of an article, we prefer not to modify. The text has been corrected (line segment 7) instead of (arrow 7). However, the quality of the figure has been improved.

Methods

Line 82 - "and 13 % "- Please, check it. 

R: The 13 % that was incorrectly left in the text has been deleted.

Line 88 (=Line 188)- "Personal conversations revealed that participants are …"- The verb tense in this section must be in the past. Please check.

R: Thank you for your comment. According to the suggestion of the other reviewer, this section has been deleted.

Line 218 – "…furthermore, the individuals' reception..." - The verb tense in this section must be in the past. Please check.

 Please check if the word "reception" should be replaced by "perception."

R:Thank you, we have changed "reception" into "perception".

Line 307 – "…the parameter tests."- Please, think about replacing "parameter" with "parametric."

R: Indeed, correctly parametric, we have corrected.

Line 354 and title of the Table - Please, consider replacing the word "foods" with "diet" to keep the consistency and precision of the terms.

R: In line 354 and in the title of the Table 'foods' has been improved with 'diet'.

Line 374 – "by the state of seclusion." - Please verify the word "seclusion" and to replace it with "reclusion."

R: The word "seclusion" has been replaced "reclusion".

Figure 3 - Figure 3 is beautiful! Ingenious and very clear explanations! Congratulations! Just as a way to contribute, for this fantastic figure, I ask if it would be possible to make arrow no.10 pass through the box of physical activity to make the mediation of physical activity well designed {the variable that enters the chain causal and contributes to the outcome}. But that is just a note. As I said, the figure is very clever.

R: Thank you for your comment. Despite our efforts, we were not able to create a more transparent diagram.

Discussion

- Very clear and precise. 

Line 482- "… measured on the „Regensburger … "– Please, to check the signal that is highlighted in red/bold.

R: Thank you, we have corrected “Regensburger” in the title of Table 4.

Reviewer 2 Report

GENERAL COMMENTS

The present study shows results of an intervention in Austrian adult women. However, the general objective is not clearly presented in the abstract and methods.

An evaluation was made at 6 months in a pandemic condition, which prevents a correct analysis of all the parameters.

In general, the document is very long and difficult to read and understand, especially the introduction. This negatively impacts the ability to "communicate" of a scientific article. Likewise, it causes the reader to be less interested in the document and may eventually be less cited.

It is recommended to shorten the document, remove the information not relevant to the study and remove many of the unnecessary details that are presented, especially in "methods".

The depth of the discussion is lower than the development given in the introduction and in other sections of the document.

It is recommended that it be done the other way around, that is, the discussion should be the most important and well-developed section in the entire document.

Line 36: please use keywords other than those used in the title

INTRODUCTION

Line 70: SOC has been explained previously, no repeat. Also, the authors say “Soc is an important determinant for health”. This need be cited.

Line 91: Figure 1 is low visual quality, please change.

Line 96: change “sense of coherence” by SOC

Line 108-112: the several consequences of sleep disruption need to be cited.  

Line 115-121: the aim please remove to final paragraph in introduction section. Also use other grammar connectors to go from SOC to talking about interventions. Write the goal for the end of everything. It is confusing to read the objectives in two places.

Line 142-143: the sentence “The theoretical background and…” it is a loose phrase and does not look good here in the end.

METHODS

Line 146: has not been explained whether women work or are at home, as that may be a bias for the SOC. Please add this information.

Line 161: try to create a flowchart on a single image, not separated or on two pages.

Line 182: correct the final of the sentence “13%”.

Line 188: “personal conversations” It is not an objective way of recording information, nor do I think it is worth mentioning in a "scientific paper".

Lines 199-211: reduce the paragraph. Please write only information about the research. Remove another detail of pandemic. This are not important.

Line 225: correct the error.

Line 228: please use only words in English.

Line 236: please use a more technical language. Change belly by abs and buttock by glutes, etc.

Line 244: you mean waist circumference? Please change.

Lines 268-278: please delete the points (…….). Change the points by “to”. it will be visually better.

RESULTS

Line 328: Table 1 is not well presented. Improve horizontal lines 1, 2, 3 and 4.  In table 1 modify "parameter" by anthropometric characteristics.

Line 331: Change “in the first table” by In Table 1.

Line 341: change “PARAMETER” by Exercises

Author Response

Response to Reviewer 2 Comments

R: We would like to thank you for reviewing this article. We have revised and modified the manuscript according to your creditable opinion.

The present study shows results of an intervention in Austrian adult women. However, the general objective is not clearly presented in the abstract and methods.

An evaluation was made at 6 months in a pandemic condition, which prevents a correct analysis of all the parameters.

R: At the time of the six-month training program, there was no epidemic in Austria.  „An evaluation was made at 6 months in a pandemic condition” Sincerely, we cannot agree with this remark.

The abstract has been clarified, the first sentence has been supplemented:

„Abstract: The original aim of this study was a follow-up assessment of a recreational program running for six months (Sept, 2019-Feb, 2020), within controlled conditions.”

In general, the document is very long and difficult to read and understand, especially the introduction. This negatively impacts the ability to "communicate" of a scientific article. Likewise, it causes the reader to be less interested in the document and may eventually be less cited.

It is recommended to shorten the document, remove the information not relevant to the study and remove many of the unnecessary details that are presented, especially in "methods".

R: In accordance with your observation, we shortened the manuscript and made efforts to make it more understandable.

It was left out from the introduction:

…Numerous studies corroborate the beneficial effects of physical activity on mental and physical health at any age [4-6]. In various physical conditions, in young people and adults alike, it is accompanied by better health characteristics, including among others, better mood [4-6]

… The SOC expressing the degree to which a person has a pervasive and dynamic, but lasting, feeling that the internal and external stimuli and stressors in his/her environment are (a) comprehensible, i.e., predictable, structured, and explicable, (b) manageable, i.e., there are resources available to meet the demands of these stimuli, and (c) meaningful, i.e., the demands are challenges worthy of engagement and coping

... Thus, a sense of context or sense of coherence (SOC) is an important determinant for health….Although the proof of SOC influencing health is not yet complete, it appears as though people who have a lower level of SOC are particularly sensitive to the hardships of life, with more of these people getting sick, and doing so more frequently [21]

….This is consistent with the results from two longitudinal studies conducted among university students in Finland and in a general population in Sweden

…. Periods of sleeplessness lasting for one month or less are called temporary insomnia, and are frequently triggered by external factors. Temporary insomnia is generally resolved when the individual adapts to the stressful event or when the events themselves are resolved.

The depth of the discussion is lower than the development given in the introduction and in other sections of the document.

It is recommended that it be done the other way around, that is, the discussion should be the most important and well-developed section in the entire document.

R: The above changes will hopefully also improve the disproportion of the chapters.

Line 36: please use keywords other than those used in the title

R: According to our opinion, the keywords may compound with the words in the title, we prefer to keep these keywords.

INTRODUCTION

Line 70: SOC has been explained previously, no repeat. Also, the authors say “Soc is an important determinant for health”. This need be cited.

R: This sentence was deleted from line 70:  “Thus, a sense of context or sense of coherence (SOC) is an important determinant for health.”. (In fact, this is the consequence of literature [13, 16])

Line 91: Figure 1 is low visual quality, please change.

R: Thank you for your suggestion, Figure 1 has been changed.

Line 96: change “sense of coherence” by SOC

R: Thank you for your comment, "sense of coherence" was replaced with SOC.

Line 108-112: the several consequences of sleep disruption need to be cited.  

R: Thank you, the section has benne cited adequately.

Line 115-121: the aim please remove to final paragraph in introduction section. Also use other grammar connectors to go from SOC to talking about interventions. Write the goal for the end of everything. It is confusing to read the objectives in two places.

R: Thank you for your comment and suggestion. The definition of the aim has been moved to the end of the introductory chapter:

Section from line 115:

„The antecedent of the research covered by this study was a follow-up study in Hungary, conducted between 2008 and 2010 [32]. In that follow-up study, 33 men and 73 women participated, it was established that compared to the start of training, a positively-oriented change occurred in the dimensions of mental, emotional, and social health as well as in all the dimensions of SOC and vegetative lability (measured with a Vegetative Lability Index) [33]. However, it is an open question, how the minimum duration required for the effects of the training to manifest, and furthermore, how enduring the achieved results would be.

     The original aim of our research started in September, 2019 was to disclose and demonstrate, in the richest possible detail, the effects of a physical recreational training program on body weight, fitness and SOC, and was planned to last for six months. The project was just about to be completed according to the research protocol when the COVID-19 pandemic took hold, halting personal contacts with the program’s participants. Due to the pandemic, we could no longer measure endurance beyond six months of the program; at the same time, we could analyze to what extent both the SOC and physical activity influence the reaction to a severe stress situation that affects everybody.

This unexpected occurrence presented an opportunity to define a new research goal, namely: examining how the research participants reacted to the pandemic and the lockdown that was implemented as a result. On the one hand, did they continue with training according to the trainer’s remote guidance, or did they exercise significantly less than before the epidemic? On the other hand, did the strength of their salutogenetic sense of coherence have an influence on this lifestyle choice, and what was most important: did all of these circumstances affect their sleep quality, as an “imprint” of the effect of stress and a sensitive indicator of health condition…

Line 142-143: the sentence “The theoretical background and…” it is a loose phrase and does not look good here in the end.

R: Line 142-143 was deleted.

METHODS

Line 146: has not been explained whether women work or are at home, as that may be a bias for the SOC. Please add this information.

R: This information is presented in line 184-187.

In section 3.2.1. we also added that " During the curfew restrictions, 60% of the employees continued their work in “home office”. "

Line 161: try to create a flowchart on a single image, not separated or on two pages.

R: Thank you for your comment, the flowchart has been presented in a single image.

Line 182: correct the final of the sentence “13%”.

R: The 13 % that was incorrectly left in the text has been deleted.

Line 188: “personal conversations” It is not an objective way of recording information, nor do I think it is worth mentioning in a "scientific paper".

R: Thank you for your comment, the paragraph was deleted.

Lines 199-211: reduce the paragraph. Please write only information about the research. Remove another detail of pandemic. This are not important.

R: We have deleted the following from line 199-211:“After that, although post offices and banks were open, citizens were required to maintain a one-meter-distance and wear a mask. Gyms were still not permitted to operate as they had pre-pandemic, and workouts could only be gradually relaunched.”

Line 225: correct the error.

R: We have corrected.

Line 228: please use only words in English.

R: The German words, whose initials make up the acronym BOGH, have been deleted.

Line 236: please use a more technical language. Change belly by abs and buttock by glutes, etc.

R: Thank you for comment. The words „belly” and „buttocks” are replaced with more relevant wording in the article.

Line 244: you mean waist circumference? Please change.

R: Thank you for your suggestion, we have corrected.

 Lines 268-278: please delete the points (…….). Change the points by “to”. it will be visually better.

R: Thank you, (……..) have been replaced with „to”.

RESULTS

Line 328: Table 1 is not well presented. Improve horizontal lines 1, 2, 3 and 4.  In table 1 modify "parameter" by anthropometric characteristics.

R: The horizontal lines in columns 5 to 7 of the table emphasize which two dates are compared with regard to the given 'anthropometric characteristics'.

„Parameter” was changed to "anthropometric characteristics" in Table 1.

Line 331: Change “in the first table” by In Table 1.

R: Thank you, “in the first table” has been changed to „In Table 1”.

Line 341: change “PARAMETER” by Exercises

R: In Table 2, „PARAMETER” has been changed to "Exercises".

Reviewer 3 Report

Overall a good job with the re-write. The paragraph that begins on line 39 needs to be re-written. Lots a mistakes in that paragraph.

Spell out numbers at the start of sentences.

Because you do not have a comparison group be very careful about making conclusions.

Author Response

Response to Reviewer 3 Comments

We would like to thank for reviewing this article. We have revised and modified the manuscript according to your suggestions.

Overall a good job with the re-write. The paragraph that begins on line 39 needs to be re-written. Lots a mistakes in that paragraph.

R: Thank you for your comment. The paragraph has been rewritten.

Spell out numbers at the start of sentences.

R: Thank you for your suggestion, we have corrected the sentences.

Because you do not have a comparison group be very careful about making conclusions.

R: The conclusion has been rewritten according to your comment:

  1. Conclusions

After perusing the scientific literature, and observing the results of these surveys, it suggests that developing areas contributing to strengthening the sense of coherence has to become one of the main directives of health developmental programs, and among these areas, physical activity has a significant role. For a female population that isn’t forced to engage in PE during their everyday life, due to their largely sedentary work, with their financial resources allowing for employing helpers to complete around-the-house work, and moreover, lacking a sufficient motivation for engaging in sports, a personalized recreational training along with parallel counselling could effectively strengthen both their physical activity and sense of coherence.

Supporting the interdependent relationship of physical activity and the sense of coherence with empirical data contributes to a complex approach to answer the question “Why are some people physically active and others not?”

Round 2

Reviewer 2 Report

Thank you to the authors for corrections.

The document improved a lot.

I'm satisfied.

Luck.

This manuscript is a resubmission of an earlier submission. The following is a list of the peer review reports and author responses from that submission.

Round 1

Reviewer 1 Report

Hello,

I wanted to open by saying thank you for conducting this study and also for being flexible with the times. It is truly fascinating that you were able to change focus and do research on the COVID pandemic. I understand these studies are never easy and I appreciate the hard work. This manuscript has valuable information that I believe can be published if you spend some time revising. The English grammar, formatting, and style are inappropriate for scientific communications. I stopped correcting for this shortly into the manuscript. It made comprehension of the material rather challenging. See my below comments:

Line 40: ‘accompanied’ has a space in the middle

Line 40-41: “accompanied by better health characteristics, among others, by better mood” is awkward English. It sounds like you just want to say ‘by better health characteristics and better mood’?

Line 46: Please define PE

Line 64-65: Decades should be ‘decade’, unless you mean the last few decades, in which case it should be quantified.

Line 66: What is a ‘sense of coherence’, is it a survey, questionnaire, etc.? Please clarify.

Line 72: Can you please clarify what you mean by ‘general resources’ and also elaborate if there is anything else that is relevant besides resisting illness? The absence of illness is not synonymous with the presence of good health – which the salutogenetic model would agree with.

Line 78: As predictors of what?

Line 78: “With a strong SOC, coping with stress works better” is unclear and informal. Do you mean to say that people with a strong SOC are able to better manage their stress? ‘Works better’ is a relative concept and would need clarification as to ‘better than what?’

Line 82 Paragraph: I am confused by this point – you say strengthening SOC in children is the most effective, but then use as an example university students (which is a completely different age group).

Line 83: ‘in young age’ is awkward English, consider ‘which has shown success in children’

Line 87: Please clarify citation format.

Line 88: Please be consistent with citation format and guidelines.

Line 88: I am confused who you are citing here – is it Antonovsky or Super? [17] is Super and does not contain Antonovsky.

Line 98: There are no arrows from SOC’s ‘Manageability’ to ‘Successful tension management’ or ‘Unsuccessful tension management’.

Line 101 Paragraph: Non-scientific quotes, informal

Line 110: Informal English

Line 110: Do you mean et al.?

Line 113: Perhaps is unclear, is it or isn’t it one of the first negative health effects, what does the research say?

Line 114-115: Please insert appropriate clarifier, i.e., inability/difficulty in starting and maintaining sleep.

Line 115: Is an interrupted sleep is the same thing as having difficulty maintaining sleep?

Line 122: What effects?

Line 123: ‘just’ is informal English.

Line 129: Grammar

Line 130-131: Please cite literature showing sleep can be an indicator of general health.

Line 132: Did the original research have anything to do with

Line 136: What is vegetative lability?

Line 144: Please verify citation format.

Line 144: Did you mean et al.?

Section 2.1: Why did you decide to choose an all female cohort and in this particular age range? Please explain.

Why did you decide to split the women into ‘younger’ and ‘older’ age groups? If you suspect an effect of age, please cite literature in the introduction.

What about the other 13%? 33+41+13 = 87

Line 171: What do you mean disposed of their leisure time? I think you mean a different word than disposed – this would imply they had 2-3 hours of free time available that they chose to remove.

STOPPED TRYING TO FIX ENGLISH GRAMMAR MISTAKES AND FORMATTING HERE – please have someone review for English grammar and proper formatting

Line 182: What measures exactly were taken?

Line 211-212: Formatting is strange, there are brackets around nothing and an extra parenthesis

Line 242-243: Why did you only do the 13 item scale at the last visit, there is no pre data to compare it to?

Line 251-252: You say that a score of 7 is ‘very often’ and that this is the highest possible score and that higher scores give a higher sense of coherence. However, you say in the same line that ‘it happens often’ is a weak sense of coherence. Please clarify.

Line 339: You say both measures were significant, however it appears that only the 6 vs. 0 healthy diet factor was significant. Please clarify.

Table 3: Given that you yourself state BMI was a poor indicator (confounded by muscle tissue gains), why did you choose to compare to BMI and not body fat %?

Line 347-348: You said only those who chose healthy foods lost weight, however, the data above indicates that 6 vs. 0 no one lost weight – their BMI’s returned to normal. Please clarify.

Line 365: Is it fair to say their physical health improved? I only see an increase in the physical variables by 1 (i.e., 16 to 17 pushups after 6 months of training). Do these small increases in physical capability translate into actual health benefits?

Figure 2: What is the large black line beneath physical activity supposed to represent? It seems to link sense of coherence 6 with ris 8, but is not an arrow.

What do the + and – indicators on the p values around physical activity indicate?

I hope you will take the time to revise the manuscript and make it more presentable. I do believe this article is publishable. I look forward to reading the update!

Reviewer 2 Report

This is an interesting study. The results of this study provided knowledge to the field of physical activity training.

I would like to suggest the authors to do some revisions.

  1. English writing warrants revision. There were several grammar errors and typos.
  2. The contents of Figure 1 were obscure and hard to read.
  3. A flow chart illustrating the recruiting participants and procedure is helpful.
  4. Why did the authors label “recreational program” as “PE”?
  5. The authors proposed that the biological and psychological effects of PE can be explained partly via epigenetic mechanisms. Did they think that the results of this study can be explained via epigenetic mechanisms? More discussion is needed.

Reviewer 3 Report

The study contains some interesting and relevant data. However, the manuscript needs a major revision and rewriting.

Introduction

The “Introduction” should be more concise. The text should be more focused on the points relevant to the current paper and the aims. Some of the issues discussed, will probably be better placed in the “Discussion” section and used to discuss and explain the observations of the current study. See further comments on the “Discussion” section below.

Materials and Methods

Measurements: SOC and RIS are based on already published questionnaires and their description in the manuscript can be limited only to any changes of what has been previously published. However, more detailed description, on the measurement tools to record “important lifestyle-related data” and the “questions related to the pandemic”, is needed. Were open or closed questions used? Were questions based on already established scales or questionnaires? How many items, etc.?

Ethics: Did the ethical committee approve the changes in the research protocol that were caused by the pandemic? Did the participants provide informed consent for these changes?

Statistics: The reason for using nonparametric tests, like Wilcoxon-rank test and Friedman test, should be explained.

Results

The Result section should be an unbiased description of the data and statistical tests, without comments or opinions from the authors. Comments and discussion on the data should be moved from the “Result” to the “Discussion” section. Examples:

  • Line 323-324: “This can probably be attributed to the fact that the body fat´s place was taken up by increased muscle mass” – move to Discussion
  • Line 361-371: “With the COVID-19 pandemic, ….. in the course of our analyses” - move to Discussion (or Introduction)
  • Line 413-414: “Presumably, … this stressful period” – move to discussion or delete.

Tables 5 and 6: OR (odd ratio) should be used for column labels instead of “Exp(B)”. Columns “B”, “S.E.”, “Wald” and “Sig.” should be deleted as they do not provide the reader with any additional information when OR and 95%CI are presented.

It is logical to assume that a positive change in BMI means a weight gain. A positive correlation (Table 3) between choosing healthy diet and BMI change would then indicate that preferring healthy foods contributes to gaining weight, which is contrary to the authors conclusion (lines 347-348). The authors should clarify this discrepancy.

Discussion

Discussion of the data in relation to other research and existing theories (described in the Introduction) is more or less nonexistent. For example, there are only two citations (line 429 and line 439) in the Discussion section. More in-depth discussion and comparison to other research - the literature - is imperative.

All data should be first presented in the result section, like the model presented in Figure 2 should be first described and referred to in the Result section. Thus, Figure 2 and a good part of the Discussion should be moved to the Result section. Furthermore, the procedure that was used to established the model presented in Figure 2 should be more clearly described. Using a Structural Equation Modelling (SEM) should be considered.

A major limitation of the original research protocol is that there is no control group included in the research design. This should be pointed out and discussed.
